# Prevalence of hepatitis B virus infection among general population of Armenia in 2021 and factors associated with it: a cross-sectional study

Anahit Demirchyan ,[1] Sandra Dudareva,[2] Serine Sahakyan,[1] Lusine Aslanyan,[1] Diana Muradyan,[1] Lusine Musheghyan,[1] Antons Mozalevskis,[3] Narina Sargsyants,[4] Gayane Ghukasyan,[5] Varduhi Petrosyan[1]

[1]Turpanjian College of Health Sciences, American University of Armenia, Yerevan, Armenia
[2]Department for Infectious Disease Epidemiology, Robert Koch Institut, Berlin, Germany
[3]Global HIV, Hepatitis and Sexually Transmitted Infections Programmes, World Health Organization, Geneva, Switzerland
[4]National Institute of Health named after academician Suren Avdalbekyan, Yerevan, Armenia
[5]World Health Organization Country Office in Armenia, Yerevan, Armenia

**Correspondence to**
Dr Anahit Demirchyan;
ademirch@aua.am

## ABSTRACT

**Objectives** This study sought to determine the prevalence and associated factors of hepatitis B virus (HBV) infection ever in life and chronic HBV infection in Armenia.

**Design** A population-based cross-sectional seroprevalence study combined with a phone survey of tested individuals.

**Setting** All administrative units of Armenia including 10 provinces and capital city Yerevan.

**Participants** The study frame was the general adult population of Armenia aged ≥18 years.

**Primary and secondary outcome measures** The participants were tested for anti-HBV core antibodies (anti-HBc) and HBV surface antigen (HBsAg) using third-generation enzyme immunoassays. In case of HBsAg positivity, HBV DNA and hepatitis D virus (HDV) RNA PCR tests were performed. Risk factors of HBV infection ever in life (anti-HBc positivity) and chronic HBV infection (HBsAg positivity) were identified through fitting logistic regression models.

**Results** The seroprevalence study included 3838 individuals 18 years and older. Of them, 90.7% (3476 individuals) responded to the phone survey. The prevalence of anti-HBc positivity was 14.1% (95% CI 13.1% to 15.2%) and HBsAg positivity 0.8% (95% CI 0.5% to 1.1%). The viral load was over 10 000 IU/mL for 7.9% of HBsAg-positive individuals. None of the participants was positive for HDV. Risk factors for HBsAg positivity included less than secondary education (aOR=6.44; 95% CI 2.2 to 19.1), current smoking (aOR=2.56; 95% CI 1.2 to 5.6), and chronic liver disease (aOR=8.44; 95% CI 3.0 to 23.7). In addition to these, risk factors for anti-HBc positivity included age (aOR=1.04; 95% CI 1.04 to 1.05), imprisonment ever in life (aOR=2.53; 95% CI 1.41 to 4.56), and poor knowledge on infectious diseases (aOR=1.32; 95% CI 1.05 to 1.67), while living in Yerevan (vs provinces) was protective (aOR=0.74; 95% CI 0.59 to 0.93).

**Conclusion** This study provided robust estimates of HBV markers among general population of Armenia. Its findings delineated the need to revise HBV testing and treatment strategies considering higher risk population groups, and improve population knowledge on HBV prevention.

## STRENGTHS AND LIMITATIONS OF THIS STUDY

⇒ Rigorous sampling methodology applied among general adult population of Armenia insured the generalisability of the study findings.
⇒ Large sample size allowed to obtain robust prevalence estimates with narrow CIs for hepatitis B virus (HBV) infection ever in life and chronic HBV infection.
⇒ Items on some important behavioural risk factors including injection drug use and risky sexual behaviours were not included in the questionnaire because of their sensitive nature.
⇒ The study methodology did not allow targeting specific population groups at high risk for HBV infection.

## INTRODUCTION

Despite the availability of effective vaccination against hepatitis B virus (HBV), it continues to be a global threat to public health, particularly in low-income and middle-income countries.[1] According to WHO 2019 data, 296 million people in the globe live with chronic hepatitis B, and 1.5 million new infections occur annually.[1] More than a million fatalities globally are attributed to HBV, which is the primary cause of morbidity and mortality from chronic hepatitis, cirrhosis and hepatocellular carcinoma (HCC), in addition to the acute infection itself.[2 3]

The commonly known risk factors for HBV are being born to an infected mother, sharing injection syringes and needles, men having sex with men (MSM), female sex workers, living with HIV and having occupational exposure to blood and blood products.[4–7] Additionally, studies have shown that sociodemographic factors including older age, male sex, low socioeconomic status, poor housing and low education could be risk factors for chronic HBV infection.[8 9]

There has been a decline in the prevalence of chronic HBV infection in many parts of the

world since a reliable vaccine became widely accessible more than 20 years ago.[10] Improved medical practices, such as screening of blood and blood products, applying safe injection techniques and implementing infection control policies further contributed to this decline.[11]

A noticeable decrease in the incidence of acute viral hepatitis infections was observed in Armenia over the last three decades. The incidence of acute HBV infection steadily decreased from 22.30 in 1990 to 0.17 in 2021 per 100 000 population.[12] One of the explanations of this decrease is the high (over 96%) coverage of hepatitis B newborn and infant vaccination introduced in Armenia in 1999.[13]

Concerning the prevalence of chronic HBV infection, no nationally representative seroprevalence studies of HBV markers were conducted among general population of Armenia before 2021. Based on synthesis of available screening data on blood donors, pregnant women, healthcare providers and patients, the best guess was that chronic HBV infection was around 2% among general population of Armenia in 2000.[13] In 2013, based on retrospective testing of frozen blood samples of over 10 000 healthy individuals aged from 1 to 60 years, the median prevalence of HBV surface antigen (HBsAg) was 1.9%–2.5%.[14 15] In the meantime, the prevalence was reported as 1.4% among blood donors, 1.5% among pregnant women, 4.3% among prisoners, 6.6% among haemodialysis patients, 7.3% among healthcare workers, 8.8% among people who inject drugs and 16.0% among patients with sexually transmitted infections.[15]

A recent review study that explored the groups of population at higher risk for chronic HBV infection in the countries of Caucasus and Central Asia stated that chronic HBV infection was higher (3.7%) among prisoners in Armenia.[16] Concerning other vulnerable groups, the estimates varied across different small-scale studies from 0% to 1.6% for MSM and from 0.3% to 0.9% for migrants (mainly seasonal workers returning from Russia).[16]

For the purpose of assessing national disease prevention and control initiatives including immunisation programmes, knowledge of the region-specific and age-specific prevalence of hepatitis B infection is essential. The study presented in this paper was the first attempt to identify the seroprevalence of HBV infection among general adult population of Armenia. The study aimed to determine the prevalence of HBV infection ever in life and chronic HBV infection in Armenia and identify the risk factors of each of these outcomes among the general adult population.

## METHODS
### Study design
This study was a cross-sectional population-based seroepidemiological study conducted throughout Armenia in May–September 2021. The study included laboratory testing of blood serum samples of the participants for biomarkers of hepatitis B, hepatitis C and SARS-CoV-2

viruses, which was shortly followed with a telephone survey among them. The study was primarily designed to measure the biomarkers for SARS-CoV-2 and the hepatitis markers were added to it. The current paper reflects the analytical approaches and findings on HBV infection.

### Sampling and sample size
The study frame was the general adult population of Armenia aged 18 years and over. We used a proportionate-to-the-population-size cluster sampling design, which included all 11 administrative units of Armenia (the 10 provinces and Yerevan city). The study participants were recruited from primary healthcare (PHC) facilities, as the coverage of population with PHC is quite high in Armenia (about 97%). We selected one-third of the polyclinics in each administrative unit using random sampling—proportionate to the served population size. Simple random sampling was then applied in the national e-health operator 'Armed' database to select the study participants from the lists of 18 years old and older individuals served by each chosen polyclinic.

The formula for estimating a proportion (n=N × X / (X + N − 1), where X=$Z_{\alpha/2}^2$ × p × (1−p)/(MOE)$^2$) of the population was used to calculate the sample size for this study. When assuming 30% prevalence of anti-SARS-CoV-2 antibodies, 1.45% margin of error (MOE), type 1 error of 0.05, and confidence level of 95%, the calculated sample size was 3832. Assuming expected HBsAg prevalence of 2% and HBV core antibody (Anti-HBc) prevalence of 15%, this sample size allowed calculation of HBsAg prevalence estimate with a MOE of 0.44% and Anti-HBc prevalence estimate with a MOE of 1.13%.

### Data collection
The 'Armed' e-health system provided the list of selected unique ID numbers to the respective polyclinic's personnel trained for contacting the study participants. The personnel contacted each potential participant by phone, following the script for the first contact, introduced the study and invited her/him to the polyclinic to take part in the study. When the participants came to the PHC facility, the research team obtained a written informed consent and collected blood samples. The participant recruitment in each polyclinic continued until the needed sample size was reached. Blood sampling was done by a trained nurse. Telephone interviews were carried out by trained interviewers within 2 weeks after the blood sampling. The structured questionnaire was administered using electronic tablets where Alchemer online tool was installed.

If the test result was positive, a trained hepatologist communicated the result to the individual and provided counselling. Negative results were sent to the participants via SMS messages.

### Serum tests and definitions
Via venipuncture, 5–10 mL venous blood was collected from each participant and centrifuged immediately for

the serum extraction. The serums were kept at 2–8°C for 7 days at most before being tested at the laboratory of the National Center for Infectious Diseases of the Republic of Armenia. This laboratory was chosen because of having the needed technical and human resources to perform third generation serological and molecular testing with the required quality for all three infections included in the study. The testing procedure for HBV included the following steps: (1) all samples were tested for total anti-HBc antibodies using third-generation enzyme immunoassay (Elecsys Anti-HBc II-Roche Diagnostics[17]); (2) if anti-HBc antibodies were detected, HBsAg test was performed (using Elecsys HBsAg-Roche Diagnostics[18]), followed by a confirmatory test if the result was questionable (within 0.9–1.0 COI); (3) if HBsAg test was positive, HBV DNA test was performed to test for the presence of HBV DNA and the viral load (using Amplisens HBV Monitor FL (detection limit 150 IU/mL)); (4) all samples positive for HBsAg were also tested for hepatitis D virus (HDV) RNA (PCR test using Amplisens HDV-FL (detection limit 100 copies/mL)). All the applied tests were WHO prequalified. The presence of antibodies against HBV core antigen (anti-HBc antibodies) was viewed as serological evidence of HBV infection ever in life, while the presence of HBsAg served as serological evidence of chronic HBV infection. (The WHO surveillance case definition for chronic HBV infection assumes the presence of biomarkers of the infection combined with the absence of signs and symptoms of acute hepatitis. We applied this definition as coming across a recent infection in a cross-sectional population study is unlikely.)

### Survey instrument and study variables

The two outcome variables of this study were anti-HBc antibody status and HBsAg status measuring HBV infection ever in life and chronic HBV infection, respectively. The survey tool was a structured questionnaire which included items related to this study grouped in socioeconomic, health status, health literacy and health behavioural domains. The sociodemographic domain included three continuous (age, family size and socioeconomic status score) and four categorical (sex, education, employment and residence) variables. The health status domain included only categorical (dichotomous) variables: hepatitis C exposure and chronic infection, several other self-reported chronic conditions (diabetes, chronic liver disease, obesity, heart disease, cancer, chronic kidney disease and asthma requiring medication), as well as symptoms experienced during the 6 months prior to the survey (vomiting, fatigue, loss of appetite, joint ache, abdominal pain, fever and nose bleeding). The health literacy domain included a communicable diseases knowledge variable measured by four questions and dichotomised at its lower third, and a score measuring the difficulty of understanding health-related information generated from responses to 10 items. The health behavioural domain included the following categorical variables: ever and current regular smoking, having tattoos, ever having been imprisoned, ever receiving blood transfusion, frequency of visiting a dentist and self-reported hepatitis B vaccination status. A variable on ever having been told by a doctor of being infected with HBV was also included in the analysis. The questionnaire is provided as an online supplemental file 1.

### Patient and public involvement statement

The survey questionnaire was pretested among eligible people and refined based on their feedback. This was the only involvement of public in this research.

### Data analysis

We used IBM SPSS Statistics for Windows, V.21 (IBM Corp.) for data management and analysis. All the analysis was weighted by sex and 5-year age groups to closely mimic age and sex distribution of the population of Armenia, as age and sex distribution in the study sample differed from that in the general population due to differences in response rates of different age and sex groups. To estimate the prevalence of HBV infection ever in life and chronic HBV infection, proportions and 95% CIs were calculated for positivity to anti-HBc and HBsAg, respectively. To descriptively analyse independent variables, frequencies and proportions were calculated for categorical variables, and means and SD for continuous variables in the total sample and in each of the outcome groups differentiated based on positivity/negativity to anti-HBc and positivity/negativity to HBsAg. Then, univariate logistic regression analysis was done. The linearity of the association of continuous variables with each outcome variable was checked on the logit scale before entering them into logistic regression analysis.[19] The final step was fitting logistic regression models to identify factors independently associated to each of the outcomes. A series of multivariable logistic regression analyses were performed with inclusion of all those variables associated with the outcome at the $p<0.25$ level in the univariate analysis.[19] The variables were entered into logistic regression analysis in different combinations using e*nter* method. The variables not related to the outcome ($p>0.10$) when controlling for the remaining variables in the model were excluded from the model, if their removal did not change the associations between the remaining variables and the outcome. The decision on keeping variables with a p value between 0.05 and 0.10 in the model was made based on their potential relevance for interpretation of the data. The final models were checked for model fit and collinearity. Cases with any missing values were excluded from the fitted models.

### RESULTS

Trained personnel from the selected polyclinics contacted a total of 9009 eligible individuals throughout the country to invite them for blood testing. Of them, 3831 (42.5%) individuals agreed and underwent blood testing. Of those tested, 3476 individuals (90.7%) participated in the phone

survey. The prevalence numbers of HBV markers were generated from the sample that was tested (n=3831), but the variables needed for the rest of the analysis (descriptive comparisons and logistic regression) were available only for those 3476 individuals who participated in the phone survey and answered the survey questionnaire. Of them, 54.5% were women and 45.5% men. The mean age of women was 46.8 years (95% CI 46.0 to 47.6), and the mean age of men 44.5 years (95% CI 43.6 to 45.3). Of men, 43.9%, and of women, 41.0% had university or higher education, while 6.3% of men and 3.0% of women had lower than secondary education. Over one-third of the participants (36.3%) were from Yerevan and 63.7% from the provinces of Armenia. Further descriptive findings and comparisons between the groups with and without HBV markers are provided in tables 1 and 2 and discussed later in the Results section.

### Prevalence of past and present HBV infection
The weighted percentage of those positive for anti-HBc antibodies indicating past or present HBV infection was 14.1% (95% CI 13.1% to 15.2%). This percentage significantly increased with age (figure 1) and was slightly higher among men than women (15.0% vs 12.2%, p=0.017).

The weighted percentage of those carrying HBsAg—the prevalence of chronic HBV infection—was 0.8% (95% CI 0.5% to 1.1%). This estimate was the highest in the age group of 30–39 years old (1.5%, 95% CI 0.8% to 2.6%) (figure 1) and was higher among men compared with women (1.1% vs 0.5%, p=0.048). The measurement of the viral load among those positive for HBsAg showed that for 61.9% of them, HBV DNA was below 150 IU/mL, for 30.2% it was between 150 IU/mL and 2000 IU/mL, and for 7.9% (n=2)—over 2000 IU/mL (in the weighted analysis). For both these cases, the viral load was 11 000 IU/mL. For those HBV DNA-positive individuals with measurable ($\geq$150 IU/mL) viral load, the median value of HBV DNA was 750 IU/mL and the IQR – 3869.6 IU/mL. None of the HBsAg-positive samples was positive for HDV RNA.

### Results of descriptive comparisons
Table 1 demonstrates the results of weighted crude comparisons of the study variables between the groups infected with HBV ever in life and those never infected with HBV.

The ever-infected group was older, with lower level of education, from a smaller-size family, and more often—residing in provinces. The proportion of employed was lower among ever infected group and the proportion of retired/disabled was higher. Anti-hepatitis C antibodies were found among the ever-infected group more frequently. The ever-infected group more frequently reported some chronic health conditions—diabetes, chronic liver disease, obesity and heart disease. However, some symptoms experienced during the last 6 months, including loss of appetite and abdominal pain, were less frequently reported by the ever-infected group. The ever-infected group demonstrated poor knowledge on

infectious diseases. The mean score of understanding health-related information was also lower among them compared with the never infected group. There were significant differences between the groups in the health behavioural domain as well. Both current and ever smokers were more prevalent among the ever infected. Higher proportion of them were ever imprisoned, and lower proportion used dental services at least once in 3–5 years. A similar negligible proportion of respondents (0.2%) in both groups reported having been vaccinated against hepatitis B.

The characteristics of those participants with and without chronic HBV infection were compared in table 2.

Those chronically infected had lower education and higher composition of men. They reported chronic liver disease much more frequently. The proportion of those demonstrating poor knowledge on infectious diseases was higher among them and the mean score of understanding health-related information was lower. Smoking was more frequent among them, and they have been imprisoned much more frequently than those free from the virus.

Among those testing positive for anti-HBc, 12 (2.7%) reported that they have ever been told by a doctor that they were infected with hepatitis B. Six of these 12 persons were also positive for HBsAg; thus, only 22.2% of those HBsAg positive were aware of their status (tables 1 and 2).

### Findings of regression analysis
The fitted logistic regression model of factors independently associated with ever being infected with HBV included the following characteristics—age, education, Yerevan versus province residence, chronic liver disease, current smoking, imprisonment ever in life and knowledge on infectious diseases (table 3).

When holding all the other variables constant, each year increase in age increased the likelihood of ever being infected with HBV by 4%. Compared with having university/higher education, having less than secondary education was associated with 1.61 times higher odds, and having secondary/vocational education with 1.26 times higher odds of ever being infected with HBV. Residing in Yerevan was protective – 0.74 times decreasing the odds of HBV infection ever in life. Suffering from chronic liver disease was associated with 2.93 times higher odds of HBV infection ever in life. Ever being imprisoned increased the odds of HBV infection ever in life 2.53 times. Current smoking was also a risk factor for ever having HBV infection, increasing its odds 1.60 times. Poor knowledge of infectious diseases was associated with 1.32 times higher odds of having HBV infection ever in life. The model had no collinearity issues with all variance inflation factor (VIF) values being below 1.13.

Several factors associated with having HBV infection ever in life, including education, chronic liver disease and current smoking, were also significant in the fitted model of determinants of chronic HBV infection (table 4).

Compared with secondary or higher education, less than secondary education increased the odds of having

**Table 1** Distribution of selected variables by participants' anti-HBV core antibody status, Armenia, 2021

| | | Anti-HBV core antibody | | | |
| | N* | Positive (n=469) | Negative (n=3005) | P value | Total (n=3474) |
|---|---|---|---|---|---|
| Socioeconomic variables: | | | | | |
| Age, mean (SD) | 3473 | 57.1 (15.7) | 44.0 (16.9) | <0.001 | 45.8 (17.3) |
| Socioeconomic status score, mean (SD) | 2696 | 46.8 (34.8) | 50.3 (35.3) | 0.075 | 49.9 (35.2) |
| Family size, mean (SD) | 3451 | 4.2 (2.1) | 4.4 (1.7) | 0.004 | 4.4 (1.8) |
| Sex | 3473 | | | 0.017 | |
| Female, % | | 49.4 | 55.3 | | 54.5 |
| Male, % | | 50.6 | 44.7 | | 45.5 |
| Education | 3456 | | | <0.001 | |
| Below secondary, % | | 7.7 | 4.0 | | 4.5 |
| Secondary/vocational, % | | 61.5 | 52.0 | | 53.3 |
| University/higher, % | | 30.8 | 44.0 | | 42.3 |
| Employment | 3438 | | | <0.001 | |
| Employed/student, % | | 41.8 | 58.6 | | 56.4 |
| Unemployed/seasonal migrant, % | | 27.2 | 28.9 | | 28.7 |
| Retired/disabled, % | | 30.9 | 12.5 | | 15.0 |
| Residence | | | | <0.001 | |
| Yerevan city, % | | 28.6 | 37.4 | | 36.3 |
| Provinces, % | | 71.4 | 62.6 | | 63.7 |
| Health status variables: | | | | | |
| Anti-hepatitis C antibodies positive, % | 3474 | 3.6 | 1.7 | 0.011 | 2.0 |
| Hepatitis C PCR test positive, % | 3474 | 1.5 | 0.6 | 0.034 | 0.7 |
| Diabetes, % | 3395 | 12.4 | 7.1 | <0.001 | 7.8 |
| Chronic liver disease, % | 3391 | 8.4 | 2.1 | <0.001 | 2.9 |
| Obesity, % | 3383 | 13.8 | 9.5 | 0.007 | 10.1 |
| Heart disease, % | 3390 | 29.0 | 12.6 | <0.001 | 14.8 |
| Cancer, % | 3394 | 2.0 | 1.5 | 0.408 | 1.5 |
| Chronic kidney disease, % | 3395 | 4.6 | 3.6 | 0.286 | 3.7 |
| Asthma requiring medication, % | 3391 | 2.0 | 2.3 | 0.864 | 2.2 |
| Symptoms experienced during the last 6 months: | | | | | |
| Vomiting, % | 3387 | 3.3 | 4.6 | 0.268 | 4.5 |
| Fatigue, % | 3391 | 41.8 | 46.0 | 0.104 | 45.4 |
| Loss of appetite, % | 3388 | 8.9 | 14.4 | 0.001 | 13.6 |
| Joint ache, % | 3393 | 37.6 | 33.4 | 0.078 | 33.9 |
| Abdominal pain, % | 3382 | 9.4 | 11.2 | 0.257 | 11.0 |
| Fever, % | 3395 | 16.5 | 25.4 | <0.001 | 24.2 |
| Nose bleed, % | 3387 | 2.9 | 4.6 | 0.108 | 4.4 |
| Health literacy variables: | | | | | |
| Poor knowledge on infectious diseases, % | 3474 | 32.8 | 24.3 | <0.001 | 25.5 |
| Health information understanding score, mean (SD) | 3005 | 52.2 (11.5) | 54.4 (9.9) | <0.001 | 54.1 (10.1) |
| Health behavioural variables: | | | | | |
| Current smoking, % | 3389 | 25.7 | 20.7 | 0.016 | 21.4 |
| Ever smoking, % | 3395 | 39.5 | 34.0 | 0.026 | 34.7 |
| Having tattoos, % | 3380 | 13.0 | 10.0 | 0.055 | 10.4 |

Continued

**Table 1** Continued

| | N* | Anti-HBV core antibody | | P value | Total (n=3474) |
| | | Positive (n=469) | Negative (n=3005) | | |
|---|---|---|---|---|---|
| Undergoing blood transfusion, % | 3380 | 4.0 | 3.4 | *0.487* | 3.5 |
| Ever having been imprisoned, % | 3387 | 4.9 | 1.2 | *<0.001* | 1.7 |
| Visiting a dentist: | 3329 | | | | |
| Once in 3–5 years/more, % | | 52.5 | 74.3 | *<0.001* | 71.4 |
| Less frequently/never, % | | 47.5 | 25.7 | | 28.6 |
| Vaccinated against hepatitis B (self-reported), % | 3397 | 0.2 | 0.2 | *1.000* | 0.2 |
| Hepatitis B told by doctor, % | 3397 | 2.7 | 0.2 | *<0.001* | 0.5 |

*Weighted estimates; *p value*: two sided.
HBV, hepatitis B virus; N, number of valid responses; PCR, Polymerase chain reaction; SD, Standard deviation.

chronic HBV infection 6.44 times. Chronic liver disease was associated with 8.44 times higher odds of having chronic HBV infection. Current smoking increased the likelihood of chronic HBV infection 2.56 times. Ever having been imprisoned was also potentially associated with having chronic HBV infection, as the observed effect range between these variables was suggestive of a weak negative to a strong positive association (aOR 3.33; 95% CI 0.81 to 13.74). Again, no collinearity issues were identified in this model with all VIF values being below 1.06.

## DISCUSSION

This is the first population-based seroprevalence study to estimate the nationwide prevalence of HBV infection in Armenia. The study demonstrated that the proportion of adult population ever infected with hepatitis B was considerably high in Armenia (14.1%; 95% CI 13.1% to 15.2%). Yet, the prevalence of chronic HBV infection was low (0.8%; 95% CI 0.5% to 1.1%), and none of the HBsAg-positive participants was positive for HDV RNA. The identified prevalence in Armenia was lower than in a study carried out in the neighbouring Georgia in 2021: the prevalence of chronic hepatitis B among adults in Georgia was 2.7% (95% CI 2.3% to 3.4%) and the proportion of persons ever infected with hepatitis B was 21.7% (95% CI 20.4% to 23.2%).[20]

Previous data for Armenia based on modelling studies showed a higher prevalence of chronic HBV infection. According to a modelling study from 2016, the HBsAg prevalence in Armenia was estimated at 1.9%,[21] and based on the Coalition for Global Hepatitis Elimination estimates, it was gradually decreasing from 3.4% in 1994/1995.[22] Surveillance data confirm that there has been a continuous reduction of new cases—mainly explained by introduction of hepatitis B vaccination among children in 1999.[13 15] This will likely contribute to further decrease in prevalence of chronic HBV infection in Armenia.[23] The prevalence in 2016 was probably an overestimation as it was relying on proxy data and not a measurement attributable to general population in Armenia. Data from screening of pregnant women in Armenia support the current finding of low prevalence. For the years 2017–2021, from over 97000 pregnant women that underwent screening, 0.66% were HBsAg positive.[24]

In crude weighted comparisons, the prevalence of both HBsAg and anti-HBc was higher in men than in women, which is similar to other studies.[20 25 26] Generally, difference in the prevalence between men and women may be related to either behavioural factors like higher sexual risk behaviours, intravenous drug use and imprisonment in men, or to biological factors.[27] Based on studies, it has been discussed that women usually develop more intense innate, humoral and cellular responses to viral infections, including hepatitis virus, than men.[27 28] After controlling for other variables, this association, however, disappeared in the model.

While for ever being infected with HBV, it was possible to detect a statistically significant increase in the prevalence with increasing age, there were no detectable differences with increasing age in the prevalence of chronic hepatitis B. However, the analysis of HBsAg prevalence by different groups was limited by low number of positive cases and, correspondingly, lack of power with the given sample size. Descriptive results, however, show that there were no chronic hepatitis B cases among those 18–29 years of age. Individuals up to age of 22 have been born after the introduction of the birth dose and childhood hepatitis B vaccination in Armenia. This potentially leads to less cases in this age group. According to the reports submitted to WHO, the impact of hepatitis B vaccination on the control of mother-to-child transmission has been demonstrated in several countries of the WHO European Region with historically intermediate and high hepatitis B endemicity, like Republic of Moldova, Georgia, Kyrgyzstan, Uzbekistan, Tajikistan and Turkmenistan.[23 29] We also did not find any HBsAg

**Table 2** Distribution of selected variables by hepatitis B surface antigen (HBsAg) status, Armenia, 2021

| | | HBsAg | | | |
| | | Positive (n=27) | Negative (n=3447) | | Total (n=3474) |
| | N* | | | P value | |
|---|---|---|---|---|---|
| Socioeconomic variables: | | | | | |
| Age, mean (SD) | 3473 | 44.7 (11.9) | 45.8 (17.4) | 0.734 | 45.8 (17.3) |
| Socioeconomic status score, mean (SD) | 2696 | 43.7 (24.3) | 49.9 (35.3) | 0.471 | 49.9 (35.2) |
| Family size, mean (SD) | 3451 | 4.2 (1.3) | 3.4 (1.8) | 0.595 | 4.4 (1.8) |
| Sex | 3473 | | | 0.048 | |
| Female, % | | 40.7 | 54.7 | | 54.5 |
| Male, % | | 65.4 | 45.3 | | 45.5 |
| Education | 3455 | | | <0.001 | |
| Below secondary, % | | 22.2 | 4.3 | | 4.5 |
| Secondary/vocational, % | | 44.4 | 53.3 | | 53.3 |
| University/higher, % | | 33.3 | 42.3 | | 42.3 |
| Employment | 3439 | | | 0.607 | |
| Employed/student, % | | 48.1 | 56.4 | | 56.4 |
| Unemployed/seasonal migrant, % | | 37.0 | 28.6 | | 28.7 |
| Retired/disabled, % | | 14.8 | 15.0 | | 15.0 |
| Residence | | | | 0.551 | |
| Yerevan city, % | | 29.6 | 36.3 | | 36.3 |
| Provinces, % | | 70.4 | 63.7 | | 63.7 |
| Health status variables: | | | | | |
| Anti-hepatitis C antibodies positive, % | 3474 | 7.4 | 1.9 | 0.097 | 2.0 |
| Hepatitis C PCR test positive, % | 3474 | 3.7 | 0.7 | 0.171 | 0.7 |
| Diabetes, % | 3396 | 11.1 | 7.8 | 0.464 | 7.8 |
| Chronic liver disease, % | 3390 | 19.2 | 2.8 | 0.001 | 2.9 |
| Obesity, % | 3384 | 14.8 | 10.0 | 0.343 | 10.1 |
| Heart disease, % | 3389 | 14.8 | 14.8 | 1.000 | 14.8 |
| Cancer, % | 3395 | 0.0 | 1.5 | 1.000 | 1.5 |
| Chronic kidney disease, % | 3395 | 3.7 | 3.7 | 1.000 | 3.7 |
| Asthma requiring medication, % | 3391 | 3.7 | 2.2 | 0.459 | 2.2 |
| Symptoms experienced during the last 6 months: | | | | | |
| Vomiting, % | 3388 | 0.0 | 4.5 | 0.632 | 4.5 |
| Fatigue, % | 3392 | 40.7 | 45.5 | 0.700 | 45.4 |
| Loss of appetite, % | 3388 | 0.0 | 13.7 | 0.042 | 13.6 |
| Joint ache, % | 3394 | 48.1 | 33.8 | 0.151 | 33.9 |
| Abdominal pain, % | 3382 | 4.2 | 11.0 | 0.508 | 11.0 |
| Fever, % | 3395 | 11.1 | 24.3 | 0.173 | 24.2 |
| Nose bleed, % | 3387 | 0.0 | 4.4 | 0.630 | 4.4 |
| Health literacy variables: | | | | | |
| Poor knowledge on infectious diseases, % | 3474 | 37.0 | 25.4 | 0.183 | 25.5 |
| Health information understanding score, mean (SD) | 3005 | 48.7 (14.4) | 54.2 (10.1) | 0.016 | 54.1 (10.1) |
| Health behavioural variables: | | | | | |
| Current smoking, % | 3390 | 44.4 | 21.2 | 0.007 | 21.4 |
| Ever smoking, % | 3394 | 53.8 | 34.6 | 0.060 | 34.7 |
| Having tattoos, % | 3379 | 19.2 | 10.3 | 0.182 | 10.4 |

Continued

**Table 2** Continued

| | N* | HBsAg | | P value | Total |
| | | Positive (n=27) | Negative (n=3447) | | (n=3474) |
|---|---|---|---|---|---|
| Undergoing blood transfusion, % | 3380 | 3.7 | 3.5 | *0.615* | 3.5 |
| Ever having been imprisoned, % | 3387 | 11.6 | 1.6 | *0.010* | 1.7 |
| Visiting a dentist | 3330 | | | *0.834* | |
| Once in 3–5 years/more, % | | 74.1 | 71.4 | | 71.4 |
| Less frequently/never, % | | 25.9 | 28.6 | | 28.6 |
| Vaccinated against hepatitis B (self-reported), % | 3398 | 0.0 | 0.2 | *1.000* | 0.2 |
| Hepatitis B told by doctor, % | 3398 | 22.2 | 0.4 | *<0.001* | 0.5 |

*Weighted estimates; *p value:* two sided.
N, number of valid responses; PCR, Polymerase chain reaction; SD, Standard deviation.

positive cases among those 70 years and older, while the percentage positive for anti-HBc was the highest in this group. On one hand, those with persistent chronic infection might have already had a lethal outcome at that age, on other hand, the sample size in this age group (n=274) might have been a limiting factor to detect prevalent cases. Furthermore, the HBsAg clearance rate with and without therapy might also influence this figure.[30] Several other factors associated with higher odds for current or past infection were identified in the multivariable regression models. For both ever being infected and chronic infection, odds were higher among people with less than secondary education and current smokers. Odds for ever being infected were higher among persons living in provinces (vs Yerevan), those demonstrating poorer knowledge on infectious diseases, and those ever imprisoned. Individuals with lower levels of education, smoking and individuals who have been imprisoned might be more likely to engage in riskier behaviour such as drug use and unprotected sexual activity leading to increased exposure to HBV.[31–33] Furthermore, lower education levels and socioeconomic disadvantages can result in reduced access to healthcare services and prevention.[34 35] Higher odds for ever being infected with HBV in provinces compared

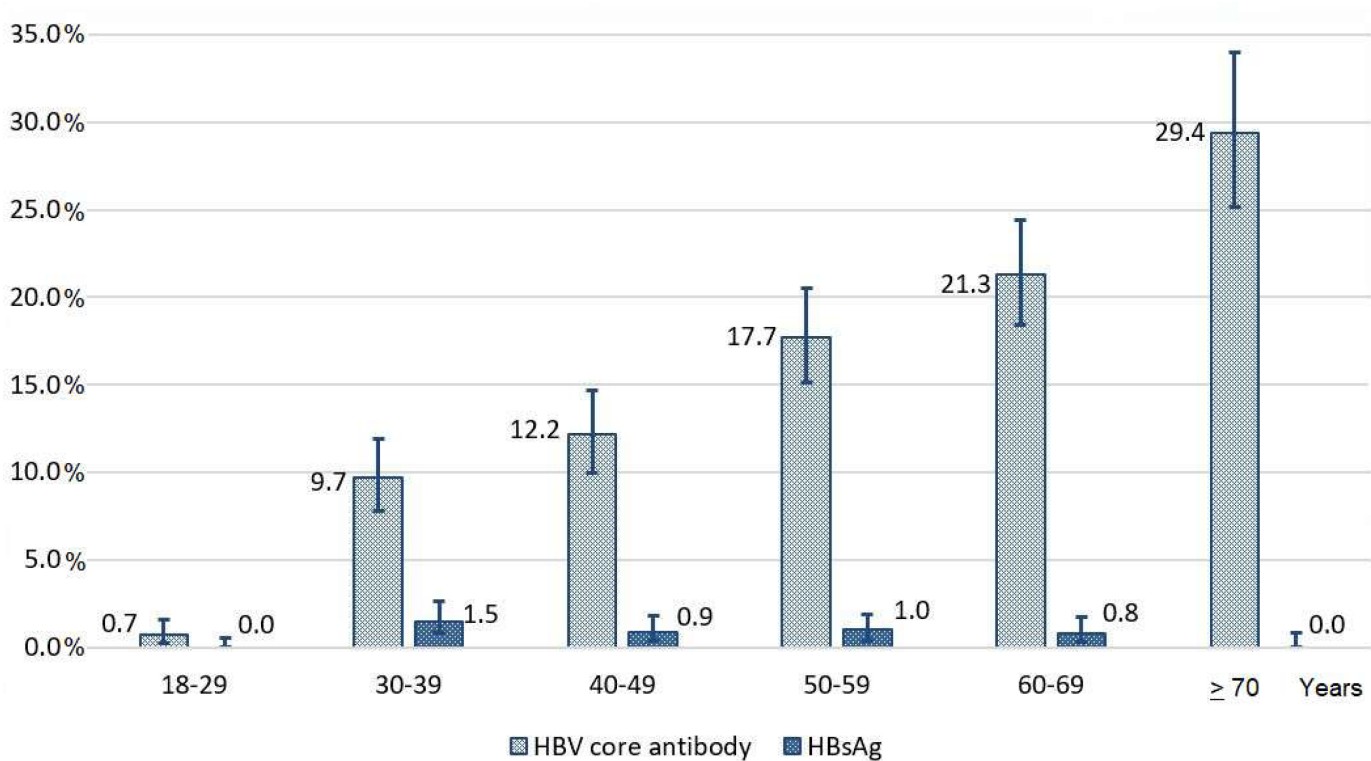

**Figure 1** Weighted prevalence estimates of hepatitis B virus (HBV) core antibody and HBV surface antigen by 10-year age groups among the study participants, Armenia, 2021.

**Table 3** Multivariable logistic regression model of determinants of HBV infection ever in life (positivity for anti-HBV core antibody) among adult population in Armenia, 2021 (valid n=3349)

| Characteristics | aOR | 95% CI | P value |
|---|---|---|---|
| Age, years | 1.04 | 1.04 to 1.05 | <0.001 |
| Education | | | |
| Less than secondary | 1.61 | 1.02 to 2.56 | 0.042 |
| Secondary/vocational | 1.26 | 1.00 to 1.59 | 0.052 |
| University/higher | 1.00 | Ref. | Ref. |
| Living in Yerevan versus provinces | 0.74 | 0.59 to 0.93 | 0.011 |
| Chronic liver disease | 2.93 | 1.88 to 4.56 | <0.001 |
| Current smoking | 1.60 | 1.25 to 2.06 | <0.001 |
| Ever imprisoned | 2.53 | 1.41 to 4.56 | 0.002 |
| Poor knowledge on infectious diseases | 1.32 | 1.05 to 1.67 | 0.018 |

aOR, adjusted OR; CI, Confidence interval; HBV, hepatitis B virus; N, number of cases.

with Yerevan could be as well related to historically poorer access to healthcare or lower level of infection prevention and control measures in regional healthcare centres, though, there is no documented evidence supporting this assumption. We could also demonstrate that poorer knowledge about infectious diseases was associated with higher odds for detection of anti-HBc, possibly explained by poorer knowledge about transmission routes, limited understanding of preventive measures, misconceptions and stigma, and poorer access to health-related information.[33 36 37]

In 7.9% of chronic infections (n=2 in the weighted analysis), the viral load was above 10000 IU/mL, indicating the need for treatment.[38] A viral load above 10000 IU/mL is a strong risk predictor of HCC, independent of HBeAg status, alanine aminotransferase (ALT) level and liver cirrhosis.[39–43] The results on viral load suggest that there is a tangible proportion of patients with chronic hepatitis B

**Table 4** Multivariable logistic regression model of determinants of chronic HBV infection (HBsAg positivity) among adult population of Armenia, 2021 (valid n=3349)

| Characteristics | aOR | 95% CI | P value |
|---|---|---|---|
| Education | | | |
| Less than secondary | 6.44 | 2.17 to 19.12 | <0.001 |
| Secondary/vocational | 1.01 | 0.42 to 2.45 | 0.984 |
| University/higher | 1.00 | Ref. | Ref. |
| Chronic liver disease | 8.44 | 3.00 to 23.72 | <0.001 |
| Current smoking | 2.56 | 1.16 to 5.65 | 0.019 |
| Ever imprisoned | 3.33 | 0.81 to 13.74 | 0.096 |

aOR, adjusted OR; CI, Confidence interval; HBsAg, hepatitis B surface antigen; HBV, hepatitis B virus; N, number of cases.

in Armenia that are currently not receiving antiviral treatment or were not aware of their infection. As the testing and treatment costs in case of a chronic HBV infection are currently mainly related to out-of-pocket payments in Armenia, the detection of cases and sustaining them in care is hindered.[24]

Based on our prevalence findings (0.8%) and given the adult population size of Armenia (~2 500 000), the estimated number of people living with chronic HBV infection in Armenia could be around 20 000, and 1600 of them (7.9% with a viral load above 10000 IU/mL) could be at high risk of HCC and in need for immediate treatment. Yet, roughly, only one-fifth of chronically infected are aware of their infection. The WHO target for 2030 is 90% of people living with hepatitis B diagnosed, which calls for improved access to testing to reach the global and regional goals.[44] Hence, there is a need to revise testing and treatment strategies in Armenia in order to find and timely treat chronic hepatitis B cases.

## Limitations

Due to limited length of the questionnaire, we were not able to include all relevant questions to determine other factors associated with higher odds of being infected with HBV. Also, as the study was conducted during the COVID-19 pandemic, some of the reported symptoms experienced within six months prior to the survey could be related to COVID-19 disease rather than participant's general health status. Furthermore, as the sample size was calculated for the SARS-CoV-2 study needs, our possibilities for calculation of the prevalence of HBV infection for different groups were limited and we might lack power to detect differences among groups and when performing regression analyses. At the same time, the sample size was appropriate to measure nationwide robust estimates with narrow CIs.

## Recommendations

In order to diagnose persons living with HBV, testing should be scaled up, especially among the population groups that are likely more affected. Improvement of state-funded preventive and testing strategies and warranting the access to state-funded hepatitis B treatment is needed in order to find and timely treat cases to prevent sequel like cirrhosis and HCC, and decrease mortality. Furthermore, to reach elimination of hepatitis B, prevention of mother-to-child transmission should be maintained at high level.[45] Validation of reaching hepatitis B control targets with a study targeting younger children (from the vaccinated cohort) could be considered in Armenia.

Also, population groups at high risk for HBV infection due to behavioural or socioeconomic reasons should be targeted with preventive measures including information campaigns, vaccination and access to needle-syringe programmes. Actions to improve knowledge on hepatitis B transmission routes, prevention and testing are needed for all population groups.

## CONCLUSIONS

This is the first nationwide representative study in Armenia providing robust estimates of anti-HBc and HBsAg prevalence among general population. While the prevalence of chronic hepatitis B is rather low in Armenia, the proportion of persons ever been infected with HBV is considerable. Age distribution of HBV infection ever in life suggests that there has been historically high incidence and that new infections possibly still occur among adults. Meanwhile, the undetectable prevalence among younger age groups suggests the positive impact of the hepatitis B vaccination.

**Acknowledgements** This work was done with technical assistance from World Health Organization (WHO) Regional Office for Europe and Robert Koch Institute. This study/research was partially funded by the WHO. The results of the research and the underlying data belong the Ministry of Health of the Republic of Armenia. The authors have been given permission to publish this article. The authors alone are responsible for the views expressed in this article and they do not necessarily represent the views, decisions or policies of the institutions with which they are affiliated.

**Contributors** AD, SS, LM, AM, NS, GG and VP conceptualised this study, and AD performed the analysis. AD, SD, LA and DM drafted the manuscript. All authors supported the interpretation of the study results and critically reviewed and revised the manuscript. AD carries the full responsibility for the study and publication and acts as its guarantor

**Funding** The data collection was partially funded by WHO Regional Office for Europe (no grant number). No funding was received for developing this manuscript.

**Disclaimer** The author is a staff member of the World Health Organization. The author alone is responsible for the views expressed in this publication and they do not necessarily represent the views, decisions or policies of the World Health Organization.

**Competing interests** None declared.

**Patient and public involvement** Patients and/or the public were involved in the design, or conduct, or reporting, or dissemination plans of this research. Refer to the Methods section for further details.

**Patient consent for publication** Not applicable.

**Ethics approval** This study involves human participants and was approved by Ethics Review Committee of the World Health Organization (Protocol ID: CERC.0013C), and Institutional Review Board of the American University of Armenia (PROTOCOL #: AUA-2021-005), as conforming to the principles embodied in the Declaration of Helsinki. Participants gave informed consent to participate in the study before taking part.

**Provenance and peer review** Not commissioned; externally peer reviewed.

**Data availability statement** Data are available upon reasonable request.

**ORCID iD**
Anahit Demirchyan http://orcid.org/0000-0002-3039-1466

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
