## [Reviewer comments · BMJ Open]

ARTICLE DETAILS

TITLE (PROVISIONAL)	Prevalence of hepatitis B virus infection among general population of Armenia in 2021 and factors associated with it: a cross-sectional study
AUTHORS	Demirchyan, Anahit; Dudareva, S.; Sahakyan, Serine; Aslanyan, Lusine; Muradyan, Diana; Musheghyan, Lusine; Mozalevskis, Antons; Sargsyants, Narina; Ghukasyan, Gayane; Petrosyan, V

VERSION 1 – REVIEW

REVIEWER	Tantuoyir, Marcarious M. Tehran University of Medical Sciences, School of Medicine
REVIEW RETURNED	12-Oct-2023

GENERAL COMMENTS	Dear authors, Your study is very important in our fight against hepatitis B virus infection especially for LMICs. The presentation of your findings is generally good though there are some concerns that need to be addressed. Kindly take your time and respond to the following comments as they will be assessed in detail; 1. In your abstract, the word 'marzes' should be replaced with a globally familiar word or phrase.2. State the p-values or percentages and their confidence intervals of the parameters mentioned in the results part of the abstract not just the latter.3. Your introduction is well-packed with information that may not all be necessary for the purpose of this study. Limit it to not more than 1.5 pages. A suggestion is to remove the information not related to adult HBV prevalence since your focus was within this population group.4. Your original data was intended for COVID-19 patients hence, the sampling was done based on SARS-CoV-2 parameters. What is the assurance that this method used did not affect the results you reported for HBV seroprevalence?5. Kindly separate your 'study design' and 'sampling and sample size' sections and you need to be concise. Additionally, the information in the methods section should be HBV-specific.6. Please provide the ethical approval code in the methods section.7. In your data analyses, You choose $p < 0.25$ as variables related to the outcome and $p \geq 0.10$ as variables not related to the outcome. What
--

	happened to the variables that fell in between this range in the logistic regression analysis? 8. Conventionally, the results section starts with the descriptive analysis of the demographic information of the study. This gives the reader a good picture of the study. I suggest you make that the first part of the results section. 9. It is observed that you used 3,474 participants in your calculations but you mentioned 3,831 in your abstract and 3,476 in the results. This can be misleading. Why the differences? 10. This statement in your discussion should be removed since it cannot be verified; "The results in Armenia are comparable to a recent study done in Ukraine, however, these data are currently not published yet." Otherwise, provide a citation for the claim. 11. You reported that "62% of the participants had HBV DNA below 150 IU/ml," indicating that Occult hepatitis B (OBI) is high in Armenia. Hence, I suggest you at least address OBI in your discussion. You may compare it with studies from other LMICs. The suggested reference could help (Reference 1). 12. The following statement in your conclusion does not reflect the aim of the paper and hence should be removed; "Furthermore, to reach elimination of HBV, prevention of mother-to-child transmission should be maintained at high level.(47)" 13. You may provide a recommendation section for your manuscript. 14. Your conclusion does not accurately reflect your work. It contains statements that were never tackled in your work. Kindly rewrite it. Kindly try to use up-to-date references, not older than 2005. Reference(s) 1. https://doi.org/10.1016/j.pmedr.2023.102401. (https://www.sciencedirect.com/science/article/pii/S2211335523002929)
--	--

REVIEWER	McNaughton, Anna University of Oxford, Nuffield Department of Medicine
REVIEW RETURNED	07-Nov-2023

GENERAL COMMENTS	Anti-HBV vaccination, should just be vaccination against HBV Strengths and limitations sections needs revising to focus more on strengths and interesting findings from the study – current it reads as a list of limitations Intro statement 'Different estimates were made at different time points' not needed. Materials and methods – what is NCID? What were the limits of detection for the HBV DNA and HDV RNA tests? Useful to know particularly for HDV RNA as no positive cases were identified. Information provided on the survey instrument and data analysis is very comprehensive and there is some duplication between the two sections – I think this could be more concise Results – HBV DNA viral load testing – was 150 IU/ml considered the limit of detection for the assay?
---

	It would be informative to present the viral load testing data as mean + IQR, rather than the way it is presented currently Was the survey of symptoms based on SARS-CoV-2 (as these samples were also collected for a SARS-CoV-2 survey), or was it adapted for viral hepatitis? It was noticeable that the anti-HBc negative group reported more symptoms than the positive group. Whilst there was a difference in family size, this was 0.2 of a person – is this confounded by a difference in rural vs urban family size at all? OR for the multivariable model should be reported as adjusted (a)OR. Discussion – “The results in Armenia are comparable to a recent study done in Ukraine, however, these data are currently not published yet.” – please either cite as a personal communication or exclude the comment, it needs evidencing. “Based on our prevalence findings, the estimated number of people living with chronic HBV infection in Armenia could be around 23,000, and 1,700 of them could be at high risk of HCC and need immediate treatment.” – some idea of how the authors calculated these numbers would be informative, and possibly more appropriate in results. Typo in final paragraph - hepatitis B transmission routes
--	--

VERSION 1 – AUTHOR RESPONSE

Reviewer: 1

Dr. Marcarious M. Tantuoyir, Tehran University of Medical Sciences

Comments to the Author:

Dear authors,

Your study is very important in our fight against hepatitis B virus infection especially for LMICs. The presentation of your findings is generally good though there are some concerns that need to be addressed. Kindly take your time and respond to the following comments as they will be assessed in detail;

1. In your abstract, the word 'marzes' should be replaced with a globally familiar word or phrase.

We replaced the word “marz” with “province” throughout the manuscript.

2. State the p-values or percentages and their confidence intervals of the parameters mentioned in the results part of the abstract not just the latter.

In the abstract, we added all the odds ratios along with their confidence intervals as suggested by the reviewer.

3. Your introduction is well-packed with information that may not all be necessary for the purpose of this study. Limit it to not more than 1.5 pages. A suggestion is to remove the information not related to adult HBV prevalence since your focus was within this population group.

We considerably reduced the Introduction, mainly deleting the parts unrelated to the HBV prevalence as suggested by the reviewer. The Introduction now is less than 2 pages.

4. Your original data was intended for COVID-19 patients, hence, the sampling was done based on SARS-CoV-2 parameters. What is the assurance that this method used did not affect the results you reported for HBV seroprevalence?

The sample size was calculated to assess the seroprevalence of antibodies against Sars-CoV-2 among adults in Armenia. Testing for hepatitis was added to use the opportunity of this nationwide sample collection. Prior the decision to include testing for hepatitis B seromarkers in the study protocol, we performed backward calculation of margin of error of the prevalence estimates of HBsAg and Anti-HBc with the estimated sample size of 3,832 (0.05 type 1 error). The expected point prevalence for HBsAg was set at 2% and for Anti-HBc at 15%. Based on the result (see table), we decided that the demonstrated precision is acceptable and will substantially increase the level of evidence on hepatitis B seroprevalence in Armenia. We now also included this information in the manuscript (the last sentence under Sampling and sample size subtitle of the Methods section).

Seromarker	Expected point prevalence	Margin of error
HBsAg	2.0	0.44
Anti-HBc	15.0	1.13

5. Kindly separate your 'study design' and 'sampling and sample size' sections and you need to be concise. Additionally, the information in the methods section should be HBV-specific.

As suggested, we separated the “Study design” and “Sampling and sample size” sections, Also, we tried to make the Methods section as HBV-specific as possible, through deleting the statements about the other two viruses to the extent possible.

6. Please provide the ethical approval code in the methods section.

We included the ethical approval codes as suggested (in the last paragraph of Methods, Data collection).

7. In your data analyses, You choose $p = 0.10$ as variables not related to the outcome. What happened to the variables that fell in between this range in the logistic regression analysis?

This was the case for only one variable in our analysis – the variable “ever imprisoned” with a p-value of 0.096. We decided to keep this variable in the fitted model of determinants of HBV infection ever in life as it is known from the literature that this is an important factor associated with HBV infection and we suspected that our study lacked power to observe a statistically significant association for it. We interpreted this finding with some caution, as the confidence level of this association was below 95%. Here is the citation of that interpretation from the text: “Ever having been imprisoned was also potentially associated with having chronic HBV infection, as the observed effect range between these variables was suggestive of a weak negative to a strong positive association (OR 3.33; 95% CI 0.81 - 13.74).” We now state the approach we applied in these cases more clearly in the Data analysis section as such: “The decision on keeping variables with a p-value between 0.05 and 0.10 in the model was made based on their potential relevance for interpretation of the data”.

8. Conventionally, the results section starts with the descriptive analysis of the demographic information of the study. This gives the reader a good picture of the study. I suggest you make that the first part of the results section.

We agree with the reviewer that the results section conventionally starts with descriptive findings. Actually, we also started with the descriptive findings, but not from the sample demographics. We started from the main outcome of the study – the prevalence of HBV infection ever in life and current HBV infection. To make this clear, we changed the former subtitle of this section (which was “Laboratory testing results”) to “Prevalence of past and present HBV infection”. We then continued with descriptive comparisons of demographic and other data between the groups with and without HBV exposure/infection. However, to address the reviewer’s comment, we added some demographic information on the study sample in the first paragraph of the Results section – before the “Prevalence of past and present HBV infection”.

9. It is observed that you used 3,474 participants in your calculations but you mentioned 3,831 in your abstract and 3,476 in the results. This can be misleading. Why the differences?

In the first paragraph of the Results section we explain this difference: “... 3831 (42.5%) individuals agreed and underwent blood testing. Of those tested, 3476 individuals (90.7%) participated in the phone survey”. In the abstract, this is also explained. Here is the citation: “The seroprevalence study included 3838 individuals 18 years and older. Of them, 90.7% responded to the phone survey.” To make this clear in the abstract, we added “(3476 individuals)” after the percentage of 90.7%. The prevalence numbers were generated from the sample that was tested – in other words – participated in the seroprevalence study (n=3831), but the variables needed for the rest of the analysis (descriptive comparisons and logistic regression) were available only for those 3,476 individuals who participated in the phone survey and answered the survey questionnaire. This is the reason for the difference between these two numbers. We described this in the manuscript (Results, first paragraph) to make it explicit.

10. This statement in your discussion should be removed since it cannot be verified; "The results in Armenia are comparable to a recent study done in Ukraine, however, these data are currently not published yet." Otherwise, provide a citation for the claim.

Since the data from Ukraine is not published yet, we removed the sentence as suggested.

11. You reported that "62% of the participants had HBV DNA below 150 IU/ml," indicating that Occult hepatitis B (OBI) is high in Armenia. Hence, I suggest you at least address OBI in your discussion. You may compare it with studies from other LMICs. The suggested reference could help (Reference 1).

Those who had low levels of HBV DNA in our sample were not those with OBI, because all of them were positive for HBsAg. “Occult HBV infection (OBI) is defined as HBV DNA detection in serum or in the liver by sensitive diagnostic tests in HBsAg negative patients...” (Zobeiri, M. (2013) Occult Hepatitis B: Clinical Viewpoint and Management. Hepatitis Research and Treatment, DOI: 10.1155/2013/259148). As described in our Methods section, under “Serum tests and definitions”, we tested for HBV DNA only those participants who tested positive for HBsAg: “...if HBsAg test was positive, HBV DNA test was performed to test for the presence of HBV DNA and the viral load”. Hence, we have no data on OBI in our sample and, therefore, we cannot discuss it in the manuscript.

12. The following statement in your conclusion does not reflect the aim of the paper and hence should be removed; "Furthermore, to reach elimination of HBV, prevention of mother-to-child transmission should be maintained at high level.(47)"

The study examined hepatitis B prevalence in Armenia. The data on the age groups demonstrated that the considerably lower prevalence among children has been reached by effective prevention of mother-to-child transmission. Therefore, we believe this recommendation remains relevant for the scope of this paper. We agree however that this is more a recommendation rather than a conclusion.

13. You may provide a recommendation section for your manuscript.

We now included a recommendation section of our manuscript.

14. Your conclusion does not accurately reflect your work. It contains statements that were never tackled in your work. Kindly rewrite it.

We agree that the recommendations that were included in the conclusions section were not directly tackled in our work. We now revised the sections accordingly.

Kindly try to use up-to-date references, not older than 2005.

We made efforts to use recent references and ended up of having only six references out of 47 that were published earlier than 2005. Indeed, 27 references out of 47 were published in/after 2015. The older references were mainly fundamental works in HBV research or biostatistics, studies in high risk populations, as well as a publication on the synthesis of local data from Armenia. We removed one of the old references, but we would prefer to keep the other five.

Reference(s)

1. <https://doi.org/10.1016/j.pmedr.2023.102401>.
(<https://www.sciencedirect.com/science/article/pii/S2211335523002929>)

Reviewer: 2

Dr. Anna McNaughton, University of Oxford

Comments to the Author:

Anti-HBV vaccination, should just be vaccination against HBV

- We changed this expression as "hepatitis B vaccination status".

Strengths and limitations sections needs revising to focus more on strengths and interesting findings from the study – current it reads as a list of limitations

- The section was completely revised.

Intro statement 'Different estimates were made at different time points' not needed.

- We removed this statement.

Materials and methods – what is NCID?

- NCID is National Center for Infectious Diseases. We replaced the abbreviation with the full name. Thank you!

What were the limits of detection for the HBV DNA and HDV RNA tests? Useful to know particularly for HDV RNA as no positive cases were identified.

- The limit for detection of HBV DNA was 150 IU/ml and the limit for detection of HDV RNA was 100 copies/ml. We added the lower limits of detection for both these tests in the manuscript, under "Serum tests and definitions" of the Methods section.

Information provided on the survey instrument and data analysis is very comprehensive and there is some duplication between the two sections – I think this could be more concise

- Thanks for pointing this out. We removed the duplicative information about the instrument domains and study variables to make the two sections more concise.

Results –

HBV DNA viral load testing – was 150 IU/ml considered the limit of detection for the assay?

- Yes, 150 IU/ml was the detection limit for HBV DNA viral load testing.

It would be informative to present the viral load testing data as mean + IQR, rather than the way it is presented currently

- To address this comment, we added the mean and IQR for those with known HBV DNA levels (above the detection limit of 150 IU/ml) in the Results: “For those HBV DNA-positive individuals with measurable (>150 IU/ml) viral load, the mean level of HBV DNA was 2787.6 (SD 4441.0) and the interquartile range 750.”

Was the survey of symptoms based on SARS-CoV-2 (as these samples were also collected for a SARS-CoV-2 survey), or was it adapted for viral hepatitis? It was noticeable that the anti-HBc negative group reported more symptoms than the positive group.

- The survey of symptoms was based on SARS-CoV-2. It was not adapted for viral hepatitis. But the list of symptoms was quite large and included also viral hepatitis-specific symptoms. Therefore, we decided to include the symptoms listed in the manuscript in our analysis.

Whilst there was a difference in family size, this was 0.2 of a person – is this confounded by a difference in rural vs urban family size at all?

- Family size was significantly different between anti-HBV core antibody-positive and negative individuals only in unadjusted analysis. This variable lost its significance after adjusting for other variables, hence, its association with the outcome was confounded by other variables, including the province vs. capital city residence. Therefore, family size was not a risk factor for exposure to HBV in this study.

OR for the multivariable model should be reported as adjusted (a)OR.

- We made the suggested change throughout the manuscript.

Discussion –

“The results in Armenia are comparable to a recent study done in Ukraine, however, these data are currently not published yet.” – please either cite as a personal communication or exclude the comment, it needs evidencing.

- We removed this statement.

“Based on our prevalence findings, the estimated number of people living with chronic HBV infection in Armenia could be around 23,000, and 1,700 of them could be at high risk of HCC and need immediate treatment.” – some idea of how the authors calculated these numbers would be informative, and possibly more appropriate in results.

- We made some additions to the cited sentence in the Discussion to make clear – how these estimates were calculated. Also, we added a sentence in the results on the percentage of those having a viral load over 10,000 IU/ml (the percentage that we used to estimate the number of those at high risk for HCC).

Typo in final paragraph - hepatitis B transmission routes

- Thanks, we corrected the typo.

With many thanks for the valuable comments and careful consideration,

VERSION 2 – REVIEW

REVIEWER	Tantuoyir, Marcarious M. Tehran University of Medical Sciences, School of Medicine
REVIEW RETURNED	13-Dec-2023

GENERAL COMMENTS	Dear authors, Your manuscript has been greatly improved. Thank you for responding to the concerns raised. However, you need to make the following minor changes; 1) Kindly present your recommendations before the conclusion of your manuscript. 2) Read through your manuscript again for minor typos. Best regards!
---

REVIEWER	McNaughton, Anna University of Oxford, Nuffield Department of Medicine
REVIEW RETURNED	18-Dec-2023

GENERAL COMMENTS	The authors have revised the manuscript considerably, making both the introduction and the materials and methods more concise, which makes the manuscript easier to follow. Having said that, there are still a few areas and points that could do with clarification - The strengths/limitations section still seems a little vague and not overly specific to the study Sampling and sample size – Some revision of the language and grammar needed throughout the manuscript – ie “We selected one-third of the polyclinics in each administrative unit via proportionate to the served population size random sampling” What is the “ARMED” database? Is this an acronym? Duplication in section explaining random sampling in this section/start of ‘data collection’. Serum tests and definitions – I don’t think the sensitivity/specificity is required for each assay given that the manufacturers are provided. Limit of detection for the PCR assays should be included though. Results – Either IQR or 95%Cis would be better for the ages of people in the study, rather than SD. Same with the viral load data. HBV DNA results jump between IU/ml and copies /ml – please make sure this is consistent. Figure 1 – needs a y-axis title so indicate the units. Assume this is prevalence (%) but the authors need to include this in the figure. Assume y. is years also? Table 1 – inclusion of the SARS-CoV-2 symptom survey. I think it is fine to include this data but it would be useful to highlight that it was intended to assess SAR-CoV-2 symptoms somewhere and it should be mentioned as a limitation somewhere. It looks odd that the not exposed group get more temperatures otherwise. Was smoking also included as a SARS-CoV-2 risk factor?
--

	Discussion – Quite lengthy still, and at times repeats the results. I feel this can still be more concise. Given the low HBV prevalence, but the continuing increase in anti-HBc prevalence, so the authors think most of the HBV exposure is occurring in adults (rather than mother to child transmission)? Some discussion of this would be useful. Recommendations – some rephrasing would be good here, ie “testing should be scaled up in first line considering population groups that are likely more affected”
--	---

VERSION 2 – AUTHOR RESPONSE

Reviewer: 1

Dr. Marcarious M. Tantuoyir, Tehran University of Medical Sciences

Comments to the Author:

Dear authors,

Your manuscript has been greatly improved. Thank you for responding to the concerns raised. However, you need to make the following minor changes;

1) Kindly present your recommendations before the conclusion of your manuscript.

- We have changed the sequence of these sections as suggested.

2) Read through your manuscript again for minor typos.

- We carefully read the manuscript to check for any minor typos. Thank you.

Best regards!

Reviewer: 2

Dr. Anna McNaughton, University of Oxford

Comments to the Author:

The authors have revised the manuscript considerably, making both the introduction and the materials and methods more concise, which makes the manuscript easier to follow. Having said that, there are still a few areas and points that could do with clarification -

The strengths/limitations section still seems a little vague and not overly specific to the study

- In this revision, we tried to make these points more specific to the study.

Sampling and sample size –

Some revision of the language and grammar needed throughout the manuscript – ie “We selected one-third of the polyclinics in each administrative unit via proportionate to the served population size random sampling”

- We revised this expression as: “We selected one-third of the polyclinics in each administrative unit using random sampling – proportionate to the served population size.” Also, we carefully checked the language of the manuscript.

What is the “ARMED” database? Is this an acronym?

- No, “Armed” is the name of the national e-health operator. We made it lowercase throughout the manuscript to avoid the confusion.

Duplication in section explaining random sampling in this section/start of ‘data collection’.

- Thanks for pointing it out. We eliminated the repetitive description from “Data Collection” section.

Serum tests and definitions – I don’t think the sensitivity/specificity is required for each assay given that the manufacturers are provided. Limit of detection for the PCR assays should be included though.

- We removed the information on sensitivity/specificity for these assays as suggested.

Results – Either IQR or 95%Cis would be better for the ages of people in the study, rather than SD. Same with the viral load data.

- We replaced the SDs with 95%CIs for the ages of male and female participants and the viral load data.

HBV DNA results jump between IU/ml and copies /ml – please make sure this is consistent.

- Thank you for this observation. We made this consistent.

Figure 1 – needs a y-axis title so indicate the units. Assume this is prevalence (%) but the authors need to include this in the figure. Assume y. is years also?

- We added the needed units to the figure.

Table 1 – inclusion of the SARS-CoV-2 symptom survey. I think it is fine to include this data but it would be useful to highlight that it was intended to assess SAR-CoV-2 symptoms somewhere and it

should be mentioned as a limitation somewhere. It looks odd that the not exposed group get more temperatures otherwise. Was smoking also included as a SARS-CoV-2 risk factor?

- The symptoms that were reported during the survey were generated as answers to a general question: “Please indicate, if you have had any of the below-mentioned symptoms in the last 6 months?” This question was asked to all participants regardless of their COVID status. The questionnaire is attached to this submission as a supplementary file. Yet, we share the concern of the reviewer that the reported symptoms could be related to COVID as the study was conducted during the COVID-19 pandemic. Therefore, we added a sentence on this as a limitation in the Discussion section: “*Also, as the study was conducted during the COVID-19 pandemic, some of the reported symptoms experienced within six months prior to the survey could be related to COVID disease rather than participant’s general health status.*”

Discussion – Quite lengthy still, and at times repeats the results. I feel this can still be more concise.

- We tried to eliminate any repetitions in the Discussion to make it more concise.

Given the low HBV prevalence, but the continuing increase in anti-HBc prevalence, so the authors think most of the HBV exposure is occurring in adults (rather than mother to child transmission)?

Some discussion of this would be useful.

- We agree with this inference, but we think that the Discussion highlights this issue sufficiently (especially given that the Discussion is lengthy already, as noted by the reviewer). Here is the citation from the Discussion on this matter: “*Individuals up to age of 22 have been born after the introduction of birth dose and childhood hepatitis B vaccination in Armenia. This potentially leads to less cases in this age group. According to the reports submitted to WHO, the impact of hepatitis B vaccination on control of mother to child transmission has been demonstrated in several countries of the WHO European Region with historically intermediate and high hepatitis B endemicity, like Republic of Moldova, Georgia, Kyrgyzstan, Uzbekistan, Tajikistan and Turkmenistan.(23, 29)*”

Recommendations – some rephrasing would be good here, ie “testing should be scaled up in first line considering population groups that are likely more affected”

- We rephrased this and some other expressions in this section.

VERSION 3 – REVIEW

REVIEWER	McNaughton, Anna University of Oxford, Nuffield Department of Medicine
REVIEW RETURNED	04-Jan-2024

GENERAL COMMENTS	The latest edits have resolved the majority of my comments. A couple of minor points - Fig 1 y axis still has no label for prevalence in the version I can see. For the section on viral load - both mean (+95%CI) and an IQR are provided. IQR is typically presented with the median value, rather than the mean. O would suggest the authors use one or the other here, rather than both.
--

VERSION 3 – AUTHOR RESPONSE

Reviewer: 2

Dr. Anna McNaughton, University of Oxford

Comments to the Author:

The latest edits have resolved the majority of my comments.

A couple of minor points - Fig 1 y axis still has no label for prevalence in the version I can see.

- Instead of a label indicating “%”, we added the “%” sign to each value on the Y axis. Figure title also explains that it depicts the prevalence estimates.

For the section on viral load - both mean (+95%CI) and an IQR are provided. IQR is typically presented with the median value, rather than the mean. I would suggest the authors use one or the other here, rather than both.

- We removed the mean and CI, and added the median value and IQR as suggested.